# Developing a Biosensor-Based Immunoassay to Detect HPV E6 Oncoprotein in the Saliva Rinse Fluid of Oral Cancer Patients

**DOI:** 10.3390/jpm12040594

**Published:** 2022-04-07

**Authors:** Chi-Sheng Cheng, Bor-Rung Ou, Feng-Di Lung

**Affiliations:** 1Department of Chemistry, Tunghai University, Taichung 407224, Taiwan; ccsheng@vghtc.gov.tw; 2Department of Stomatology, Taichung Veterans General Hospital, Taichung 407219, Taiwan; 3Department of Animal Science and Biotechnology, Tunghai University, Taichung 407224, Taiwan; brou@thu.edu.tw

**Keywords:** biosensor, E6 oncoprotein, human papillomavirus, oral cancer, solid phase peptide synthesis, salivary biomarkers, surface plasmon resonance

## Abstract

Transmission of Human papillomavirus (HPVs) is faithfully associated with carcinogenesis of oral cavity and oropharyngeal cancers. Therefore, clinical researchers may need to generate customized antibodies for the upcoming ELISA-based analysis to discover rare but valuable biomarkers. The aim of study was to develop and generate a biosensor-based immunoassay for early screening HPV-related oral cancer via saliva rinse fluid analysis. A peptide fragment of high-risk HPV subtype 16/18 protein, E6 protein (HP-1 protein sequence 48–66), was designed and synthesized, followed by the generation of polyclonal antibodies (anti-HP1 IgY) in our university-based laboratories. The titer and specificity of antibodies were determined by enzyme-linked immunosorbent assay (ELISA), and the Surface Plasmon Resonance (SPR) biosensor-based method was developed. Kinetic analyses by SPR confirmed that this designed peptide showed a high affinity with its generated polyclonal antibodies. Saliva fluid samples of thirty oral cancer patients and 13 healthy subjects were analyzed. SPR indicated that 26.8% of oral cancer patients had higher resonance unit (ΔRU) values than normal subjects. In conclusion, we developed a biosensor-based immunoassay to detect HPV E6 oncoprotein in the saliva rinse fluid for early screening and discrimination of HPV-related oral cancer patients.

## 1. Introduction

The development of new tools of biofluid technology for biomarker detection in saliva is challenging. However, saliva-based examination is potentially useful for oral cancer screening, monitoring treatment response and developing personalized medicines. Two-dimensional polyacrylamide gel electrophoresis (2-D PAGE) and Mass Spectrometry based assays may not be suitable for application in clinical settings because of analysis needed in laboratory-based facilities and it being time-consuming for data interpretation [1,2]. Therefore, it is urgent to develop more practical detection approaches of salivary biomarkers such as “Lab-on-the-Chip” or “Point-of-care testing” (POCT) [3,4,5]. Thus, some potential alternative techniques were introduced to establish better methods for detecting salivary biomarkers with non-invasive, efficient, and convenient features for clinical practice [6]. Surface Plasmon Resonance (SPR) is considered a potentially promising technique. Currently, it is an attractive biosensor approach using antibody-based immunoassay [7,8,9].

Growing evidence supports the role of HPV in the cause and response of oral cancers squamous cell carcinoma (OCSCC) and oropharynx squamous cell carcinoma (OPSCC) [10]. Latent viruses with persistent infection could be involved in the malignant transformation of OCSCC. For example, the long-term presence of such viruses in basal cells of tonsil or gingival tissues might be the reservoir for HPV in the oral cavity [11]. Additionally, HPV related oropharyngeal cancer increases the incidence steadily [12] and reveals a favorable outcome after radiotherapy treatments [13]. HPV vaccines targeting HPV 16 and 18 are effective against OPSCC [14].

Clinical researchers may need to generate customized antibodies for the upcoming ELISA-based analysis at a university-based laboratory to discover rare but valuable biomarkers. Here, we present a streamlined procedure to develop the biosensor approach of the antibody-based immunoassay using the SPR technique. First, we designed and synthesized an antigen, a novel peptide fragment (protein sequence 48–66; HP-1) of HPV 16 E6 protein [15]. Chickens were immunized to produce anti-HP-1 IgY for the biosensor-based assay. To determine the affinity between anti-HP-1 IgY and targets of HPV-related E6 protein, SPR was used for the measurement which was used to detect the E6 protein in saliva of oral cancer patients. Secondly, Saliva fluid samples of thirty oral cancer patients and 13 healthy subjects were analyzed (Figure 1).

## 2. Materials and Methods

Our study was approved by the Committee on Human Research of the Veterans General Hospital (IRB approval #:SE13339b#1). All participants provided written informed consent for the collection of samples and analysis. Saliva samples were collected from 30 oral cancer patients and 13 healthy subjects at the Department of Stomatology, Taichung Veterans General Hospital. For healthy subjects, inclusion criteria were aged > 18 years and with no history of neoplasm. From 2015 to 2017, we enrolled 30 advanced T3–4 oral cancer patients who were never treated according to the 7th AJCC on Cancer TNM staging system. The collected saliva was added with an enzyme inhibitor (Protease Cocktail Inhibitor; cOmplete Tablets, Mini, EASYpack, Roche, Switzerland). The collection steps were as follows: 20 to 30 mL of sterile distilled water were used to rinse the oral cavity gently and discarded. Another 5 mL distilled water was used for 1-min gargling and then spitted to collect all of the saliva, and samples were placed in ice immediately. After adding the protease inhibitor, the collected saliva sample was centrifuged at 3000 rpm at 4 °C. The supernatant of the sample was extracted and stored at −80 °C.

We designed and synthesized a peptide fragment E6 48–66 in HPV 16 E6 protein (named as HP-1 peptide) and used that as an antigen for immunization in chickens. The egg yolk polyclonal antibodies were integrated into the immunoassay to develop a biosensor using the surface plasmon resonance. HP-1 was modified and extended in length at peptide region 52 to region 62. This region is well documented and is a target for CD4 T-cell epitope [15]. Solid-phase peptide synthesis (SPPS) was used for the HP-1 peptide. The amino acid protecting group of the synthesized peptide was removed through the AM resin with a 95% trifluoroacetic acid (TFA) aqueous solution. The peptide was then cleaved from the resin to yield the crude HP-1 peptide. Afterwards, the HP-1 peptide was separated and purified with RP-HPLC, and the peptide was identified with MALDI-TOF mass spectrometry (MALDI-TOF MS).

Immunization procedures followed the terms of the Institutional Animal Care and Use Committee (IACUC). The immune response in Lai Hen chickens was that the antigen was boosted by a complete adjuvant (complete Freund’s adjuvant, CFA). The adjuvant prevents rapid disintegration of the antigen, prolonging the antigenic exposure to the immune system and enhancing the response of immune cells. The immunized injection contained an emulsified mixture of 300 μg formed by 150 μg of antigen and 150 μL of complete adjuvant (in 1:1 ratio). The second injection was given 14 days later, and consecutive immunizations were performed at 3-week intervals until reaching a stable sera titer. At each boost injection, the serum titer was checked, and the animal with the highest titer was selected, and her eggs were collected for antigen separation. 

The liquid phase extraction separated antibodies from the egg yolk, following the method described by Polson [16]. First, 15 mL egg yolk was extracted from an egg and mixed with distilled water in a 50 mL centrifuge tube. Then, 15 mL of chloroform was added, followed by a slow and even shaking. Finally, the supernatant was extracted by centrifugation at 4000 rpm for 5 min. The supernatant contained the antibodies, which were stored at −20 °C. Lastly, the Smith assay determined the concentrations of chicken anti-HP-1 IgY antibodies in the egg yolk. The standard protein was bovine serum albumin.

Titers of egg yolk anti-HP-1 IgY antibodies were determined with the indirect enzyme-linked immunosorbent assay (ELISA). Briefly, into microtiter wells on the ELISA plate, 100 μL of each HP-1 peptide standard and 100 mL chicken anti-HP-1 IgY antibody were pipetted. Dispensed into each well was 100 μL of goat anti-chicken-HRP conjugate reagent. The mixture was allowed to incubate at room temperature under shaking (750 rpm) for 90 min. Next, the incubation mixture was removed by flicking plate contents into a waste container. Wells were rinsed and flicked five times with wash buffer, and the plate was struck sharply onto paper towels to remove residual water. Then, the TMB reagent (100 μL) was added to each well, gently mixed for 10 s, and allowed to incubate at room temperature in the dark with mechanical shaking (750 rpm) for 20 min. Next, hydrochloric acid (1 N, 100 μL) was added to each well and gently mixed for ~30 s to stop the enzymatic reaction (making sure that the blue color had changed to yellow completely). Within 15 min, the 450 nm absorbance was measured for each well using a VERSAmaxTM microplate reader (Molecular Devices, Boston, MA, USA).

The Kinetic/Affinity analysis was performed using the Biacore 3000 to study the kinetics (binding and dissociation rate) and affinity (binding strength) of interactions of the antigen–antibody complexes. The following experiments adopted a multi-cycle mode, in which the regeneration step was performed after dissociation of each injection sample to ensure similar conditions of chips before each sample injection. The following experiment aimed to investigate the interactions between anti-HP-1 IgY antibody and E6 protein (commercially purchased). Anti-HP-1 IgY antibodies diluted in serial concentrations were used as analytes. The highest concentration was 10.00 μg/mL, and from that diluted serially to 5.00 μg/mL, 3.33 mg/mL, 2.50 mg/mL, 2.00 mg/mL, and 1.66 μg/mL. Induction spectra were detected with the E6 protein immobilized on the biosensor chip.

Initially, EDC/NHS was used to activate the surface of the sensor chip at a final concentration of 0.2 μg/mL, and anti-HP- 1 polyclonal antibody were covalently bound to the surface of the chip (first tested with sodium acetate solution at pH of 3.5, 4.0, 4.5, 5.0, 5.5 optimal bonding conditions, and then repeated at pH 3.5). The final EA masked the remaining activating sites. A sensorgram was generated by reacting a 1⁄2 diluted saliva sample from the 43 subjects as analytes. Anti-HP-1 antibodies (IgY) were immobilized on the sensor chip, and the saliva sample was tested in the solution that flew through the channel on the chip for monitoring the to-be-detected interactions. The sample mixture was diluted with an HBS buffer and analyzed with Biacore 3000 at a flow rate of 20 μL/min.

## 3. Results

### 3.1. Synthesis, Purification, and Titer Determination of HP-1 Peptide

#### 3.1.1. Solid Phase Synthesis of the HP-1 Peptide and Purification

The peptide sequence of HP-1 was HN2-EVYDFAFRDLCIVYRDGNP-COONH2, and the theoretical molecular weight was 2292.08 Da. RP-HPLC purified the crude peptide mixture, and the retention time (Rt) of the HP-1 peptide was determined as 14.73 min. The solution collected at 14.73 min was freeze-dried, and its molecular mass was determined as 2293.5 m/z by MALDI-TOF MS. These results indicated the successful synthesis and purification of HP-1 peptide (Figure 2a,b).

#### 3.1.2. Determination of the Titer of Chicken Anti-HP-1 Antibody by Indirect ELISA

The highest concentration of egg yolk antibody was 250 μg/mL, which was diluted serially to 125 μg/mL, 62.5 μg/mL, 31.25 μg/mL, 15.625 μg/mL, 7.8125 μg/mL, and 3.90625 μg/mL (Figure 3).

The detected responses of egg yolk antibodies ranged from 0.25 mg/mL to 0.0625 μg/mL were found to be dose-dependent. In addition, we found an excellent linear relationship between the antibody and antigen was within the concentration of 1.953125 μg/mL. Therefore, in the following ELISA analysis, the antibody concentration was set up at 125 μg/mL.

#### 3.1.3. Analyses of Kinetics and Affinity of E6 Protein and Anti-HP-1 Polyclonal Antibodies

E6 protein was immobilized on the sensor chip (Figure 4a). Anti-HP-1 IgY antibodies had led to changes in RU. ΔRU also tended to increase with the injected concentration of anti-HP-1 multi-strain antibody (Figure 4b). Using the “1:1 binding” model for kinetic analysis, the binding rate (ka) and dissociation rate constant (Kd) calculated were 3.406 × 102 M^−1^ s^−1^ and 1.191 × 10^−4^ s^−1^, respectively (Table 1). When the chip flowed through the anti-HP-1 polyclonal antibodies at various concentrations, the highest ΔRU value achieved was plotted against its concentration. The equilibrium dissociation constant (KD) was calculated to be 1.753 × 10^−7^ M (Table 2). The kinetic and affinity analysis ensured a high specificity antibody of the generated anti-HP-1 IgY for detecting biomarkers in the immunoassay.

#### 3.1.4. Detecting E6 Protein in Saliva of Oral Cancer and Normal Subjects

Initially, the optimal pH was determined by the second channel of the CM5 sensor chip to determine the optimal pH; the maximal immobilization of antibodies was found at pH 5.0, with a ΔRU value of 24,000 RU. Samples from a total of 43 subjects (N: 13 normal subjects and P: 30 patients III + IV) were tested using half diluted saliva as analytes, in which anti-HP-1 IgY antibodies had shown immobilization at the sensor chip. The sensorgram revealed that some patients’ saliva had interacted more strongly with antibodies. This finding indicated that these few patients had E6 protein present in their saliva samples. When the value of RU was set at 18.694, we found both positive and negative results (means: −13.102; standard deviation: 12.324 RU; the range of confidence interval: −44.898 to 18.694). With the upper limit as the cutoff level, it was found that all normal subjects with values < 18.694 RU were classified as negative results, and 26.7% of oral cancer patients (stage III + IV) showed positive results (Figure 5).

## 4. Discussion

In this study, we designed and synthesized a specific candidate peptide, HP-1, for HPV E6 protein as a salivary biomarker for oral cancer. The kinetic analysis demonstrated a high affinity between our peptide immunized IgY antibody and the E6 protein. Thus, we anticipate that this self-generated IgY can provide a fundamental immunoassay model to develop tools for detecting salivary biomarkers at university-based laboratories, particularly for the rare but valuable peptides as candidate biomarkers.

In addition, we successfully established a biosensor/biochip immunoassay to detect the presence of HPV E6 protein in the saliva of oral cancer patients. Empirically, we analyzed 30–40 samples for testing our biomarkers and biosensor in this study. Although the small sample size can lead to a wide range of confidence intervals and inaccuracies in statistical hypotheses testing, it showed that 26.8% of advanced oral cancer patients had higher levels of E6 protein in their salivary samples when considered at a cutoff level of 18.694 ΔRU. Following this result, one-fourth of the HPV E6 positive rate is compatible with previous reports of incidence of HPV-related oral and oropharyngeal cancer patients [17,18]. The further confirmation study will precede large sample size numbers in the future.

The scheme of detecting salivary biomarkers for oral cancer is a promising step forward for individualized medicine. Bio-fluids, such as saliva, could contain tumor-related DNAs and proteins related to the diseases [19,20]. Several studies have reported that HPV DNA in the saliva of oral cancer patients could be used in a non-invasive means for assessing HPV tumor status [21,22]. For example, HPV-16/18 status of saliva and plasma is likely a screening tool for HPV-16/18-positive patients and a predicting tool for post-treatment three-year recurrence in oral cancer patients [23].

However, targeting HPV oncoproteins, as an oral cancer biomarker facilitates new approaches in salivary-based assays for early diagnosis, clinical staging, and prediction of treatment outcomes [24]. In the early stage of HPV infection, the virus first enters the upper stratified squamous epithelial cells and reproduces by multiplying in host cells to extend its life cycle. After reproducing the E6 and E7 viral proteins, a host cell loses its ability to inhibit its cell division and the normal restriction of cell differentiation. When E7 binds to pRb (retinoblastoma tumor suppressor protein), it allows E2F to dissociate and guide host cells to start protein expression required for DNA replication. Moreover, E6 inhibits p53 activities, which played a role in DNA repair, leading to apoptosis. If HPV infected DNA-damaged cells due to dysfunctions of pRb and p53, they could continue to replicate and ultimately into the malignant transformation processes [25,26].

Compared to HPV DNA, E6/E7 protein expression might be more directly associated with human cancer risk. The reason might be that the presence of HPV DNA is not enough to prove the causality of malignant transformation. Mainly HPV E6/E7 oncogene expression is considered necessary for carcinogenesis. Therefore, the lack of HPV E6/E7 oncogene expression represents the carcinogenic causality of HPV does not exist [27]. Based on this assumption, the screening threshold of E6/E7 oncoprotein should be quantified and used to detect HPV-related cancer. In 2018, Zhang tested the feasibility of human papillomavirus E6 and E7 oncoproteins as the target to diagnose cervical cancer. This study concluded that using E6/E7 assay can help diagnose the early stage of cervical cancer. Compared to the ThinPrep cytologic test (TCT) and HC2 HPV DNA test (HC2), E6/E7 assay, Western blot assay showed that it is important to diagnose CIN2+ with the moderate capability of sensitivity and specificity by a positive threshold value of 20%, which can apply in early diagnosis with cancer screening [28].

Currently, there are several tests available for diagnosing HPV-related OPSCC. HPV DNA and mRNA assess HPV-related infection or tumors by PCR and Rt-PCR in the paraffin-fixed tissue specimens. The main strength of those examinations is high sensitivity and specificity; however, they are unpractical for usage in clinical applications [29]. The alternated method is testing p16 by immunohistochemistry as a surrogate HPV-positive biomarker. Although testing p16 by IHC is the most common method used to diagnose clinically but is still recognized to have a 20% false-positive rate in HPV-negative patients. Furthermore, the diagnosis can be subjective due to the lack of a standard scoring system for differentiating HPV positive on IHC testing [30]. Regarding development of a simple and rapid test for diagnosing HPV status in HNSCC patients, detecting E6 protein for screening or diagnosis of HPV-related oral cancer by saliva fluid. Comparisons of detection and diagnosis methods of HPV in oral cancer patients are summarized in Table 3.

Application of IgY is of potential use for treatment and cancer-related research and in developing immunotherapy against surface tumor markers in cancer cells. Several studies reported on the capacity of IgY-based immunoassay for diagnostic purposes and research [31,32]. This has an essential advantage over mammalian IgG, regarding the lack of a false-positive due to them not being a crossed reaction with the rheumatoid factor. The IgY antibody provides similar reactivity to the mouse monoclonal IgG antibody in mice. Furthermore, since only eggs are collected, and no animals are sacrificed, the approach raises little moral controversy. Regarding the efficiency of antibody production, IgY antibodies from an egg can be produced as high as 10 mg/egg. In addition, housing hens is inexpensive, and affinity purification is easy to do and fast. Due to these advantages, it is interesting that avian IgY replacing mammalian IgG is more likely preferred as a diagnostic tool in ELISA assay for infectious diseases and biomarkers of cancer. For example, IgY antibodies are used in sandwich ELISA to detect breast cancer biomarkers, namely the CA 15-3 antigen in the serum [33]. Although IgY antibodies are commonly used in ELISA and immunoblotting, some new detection systems have been developed, such as fluorescence immunoassay, surface plasmon resonance (SPR) [34], and immunochromatography [35], as well as in developing immunotherapy against surface tumor markers in cancer cells [36].

Surface plasmon resonance (SPR) techniques have a high potential for clinical applications to obtain a more efficient salivary-based analysis. The primary measuring mechanism of SPR immunoassay is detecting changes in the optically refractive index of the metal chip based on interactions between antigen and antibody that immobilize products in samples. SPR showed some advantages in proteomic analysis [37]. This SPR technique can be used to measure the kinetics of interaction, with acknowledged specificity and affinity between the antibody and the target antigen [38]. SPR, compared with conventional assays, is a time-efficient method and detects biomarkers without involving labeling. After standard preparation procedures, antibodies can be applied in SPR with immobilization on the biosensor chip. We postulated an efficient assay to identify salivary biomarkers using SPR. This saliva biomarker assay method is inexpensive and has convenient pipeline procedures, which require only an hour to finish also reducing labor and cost investments to process immunoassay [39]. It is potentially applicable in clinical settings which require testing of samples with quick, chair-side, and convenient analysis procedures [40]. It also can apply to other anatomic sites, such as the cervix, to screen and test HPV-related cancer.

## 5. Conclusions

We reported a biosensor/biochip-based immunoassay for identifying saliva biomarkers in oral cancer patients. It is potentially helpful for oral cancer screening, monitoring treatment response, and developing personalized medicines. Furthermore, such an approach of IgY-based internal detection is also suitable for biomarker detection of new pandemic viruses (such as COVID-19), as the method is cost-effective and is fast in implementation and analysis. The primary goal is to develop a biosensor-based assay to detect salivary biomarkers of oral cancer patients, a rapid, efficient, and non-invasive diagnostic tool.

## 6. Patents

This manuscript builds on our study to identify salivary biomarkers based on immunoassay biosensor/biochip techniques. The U.S. patent (US 10379123) of this technique has been filed.

## Figures and Tables

**Figure 1 jpm-12-00594-f001:**
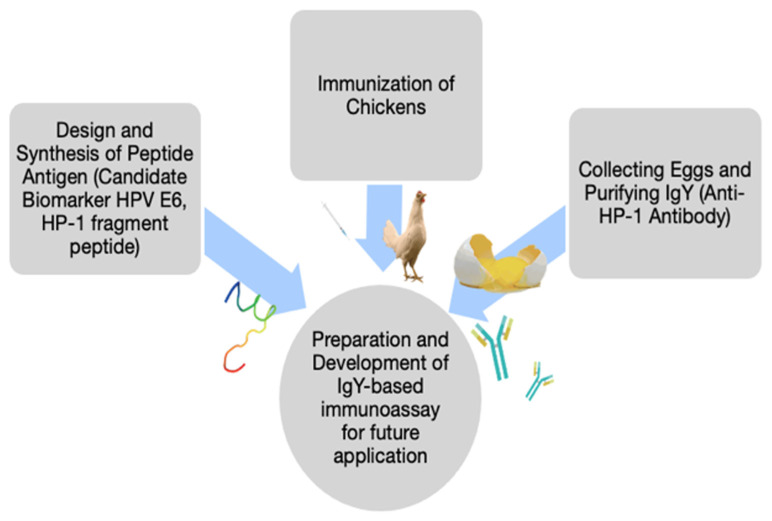
The schematic diagram of synthesis, purification, and analysis of Chicken anti-HP1 IgY.

**Figure 2 jpm-12-00594-f002:**
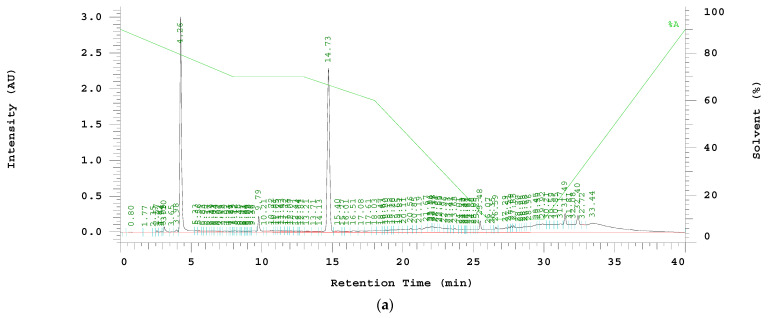
(**a**) RP-HPLC chromatography of HP-1 peptide. The gradient started at 10–90% acetonitrile (contain 0.05% TFA), and continued 48–49% acetonitrile from 10 min to 20 min, then 90–10% acetonitrile over 14 min at a flow rate of 4 mL/min. The retention time of target peptide is 14.73 min. (**b**) HP-1 pure peptide characterized by MALDI-TOF; the major peak of target peptide is 2293.5 Da.

**Figure 3 jpm-12-00594-f003:**
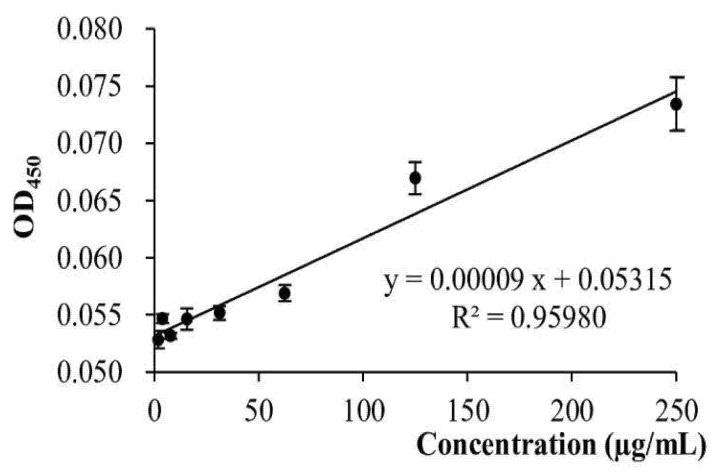
Detection range of the antibody and antigen reaction showing a good linear relationship. Consequently, the ELISA analysis that followed had adopted an antibody concentration at 125 μg/mL.

**Figure 4 jpm-12-00594-f004:**
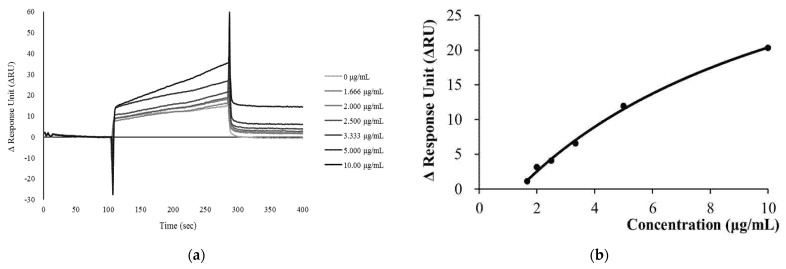
Analyses of kinetics and affinity of E6 protein and anti-HP-1 polyclonal antibodies. (**a**) The SPR Sensorgram showed that immobilized E6 proteins on the sensor chip and variant concentrations of anti-HP-1 IgY antibodies. (**b**) The analytes were flowed the chip through the anti-HP-1 polyclonal antibodies at various concentrations; the highest ΔRU achieved was plotted against its concentration.

**Figure 5 jpm-12-00594-f005:**
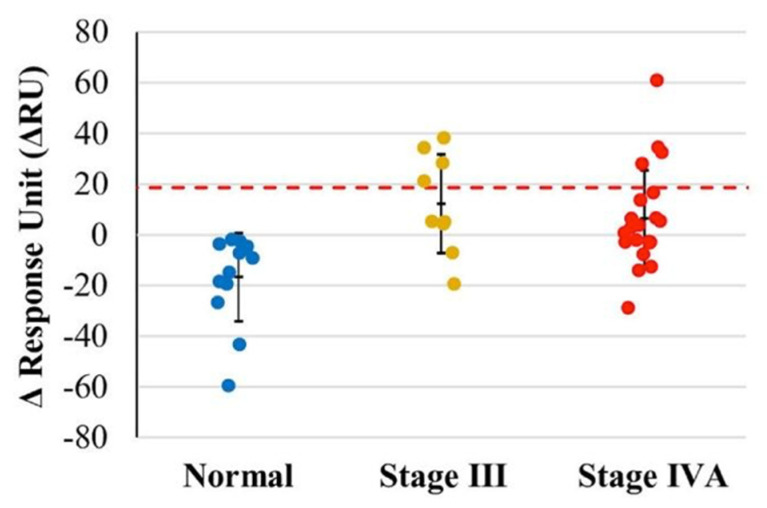
Analysis of normal and oral cancer patients of E6 with anti-HP-1 IgY antibodies. ΔRU values of normal, stage III and stage IV represented by the colors blue, yellow, and red. A total of 26.7% of oral cancer patients (stage III + IV) showed positive results.

**Table 1 jpm-12-00594-t001:** Kinetic analysis of immobilized of E6 protein and anti-HP-1 IgY antibodies with different concentrations.

k_a_ (1/Ms)	k_d_ (1/s)	K_D_ (M)	Rmax (RU)	tc	Chi^2^ (RU^2^)
340.6	1.191 × 10^−4^	3.497 × 10^−7^	1385	8.020 × 10^6^	0.524

**Table 2 jpm-12-00594-t002:** Affinity analysis of immobilized of E6 protein and anti-HP-1 IgY antibodies with different concentrations.

K_D_ (M)	1.753 × 10^−7^
Rmax (RU)	52.94
offset (RU)	−6.557
Chi^2^ (RU^2^)	0.288

**Table 3 jpm-12-00594-t003:** Summarization of detection and diagnosis of HPV in oral cancer.

Source of Samples	Target Biomarker	Methods	Sensitivity	Specificity	Advantages	Disadvantages
Histopathological specimen [40]	HPV-DNA	polymerase chain reaction (PCR)	>90%	>90%	High assay sensitivity and specificity	Not directly indicated that cancer is induced by HPV
	HPV-DNA	in situ hybridization	>90%	>90%	High agreement with HPV DNA assays	Not suitable for clinical application
	p16INK4A	Immunohistochemistry	>80%	>80%	Reliable indicator biomarker	High false positive rate, and no standard scoring system available
Serum [28,41]	HPV antibody	E6/E7 antibody analysis	>80%	>90%	Indicates the presence of HPV-associated cancer	At any anatomic site, not specific in HNSCC
	HPV E6/E7 oncoproteins	Western blot analysis	>65%	>30%	Indicates the presence of HPV-associated cancer	At any anatomic site and low sensitivity of assay
Saliva [42,43]	HPV DNA	polymerase chain reaction (PCR)	>90%	>70%	High assay sensitivity and specificity	Single time-point measure of HPV exposure and not directly indicated that cancer induced by HPV
	HPV E6/E7 transcripts	qRT-PCR	>90%	>95%	Achievable in laboratories	Limited utility in clinical practice

## Data Availability

Not applicable.

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
