# Peer review of "Developing a Biosensor-Based Immunoassay to Detect HPV E6 Oncoprotein in the Saliva Rinse Fluid of Oral Cancer Patients"

_jpm, 2022, doi:10.3390/jpm12040594_

Round 1

Reviewer 1 Report

The authors present an own patented biosensor/biochip immunoassay for detection of HPV in saliva of H&N carcinoma patients. It is an important issue to have a non-invasive rapid tests that can detect malignancy or its follow up and treatment.

The method is nicely described.

I would like to hear more on comparison with available assays: sensitivity, specificity, PPV, NPV etc. What are main advantages and disadvantges. table comparsion with available assyas would be useful.

Any possibility for use in cervical cancer screening?

Author Response

Thanks for your recommendation.  I have added our opinions in the section of discussion, in page 8 (sentences #264-276), and make a Table. 3 to replay to questions.

Reviewer 2 Report

The authors described "Developing a Biosensor-Based Immunoassay to Detect HPV E6 Oncoprotein in the Saliva Rinse Fluid of Oral Cancer Patients". In this study, they had designed and synthesized a specific candidate peptide, HP-1, for HPV E6 protein as a salivary biomarker for oral cancer.  The kinetic analysis demonstrated a high affinity between their peptide immunized IgY antibody and the E6 protein. In addition, analysis of normal and oral cancer patients of E6 with anti-HP-1 IgY antibodies. 26.7% of oral cancer patients (stage III+IV) showed positive. It may be potentially helpful for oral cancer screening, monitoring treatment response, and developing personalized medicines. I have one question. How did you decide the sample size? Please add the calculation method.

Author Response

Before this project, we did research by similar study approaches, which explored possible targets in the saliva of oral cancer patients by SPR. Target biomarkers included p16 protein, EGFR, ZNF510 fragment peptide, and commercially purchased E6 and E7 oncoproteins and including this HP-1 peptide. After evaluating 16-20 subjects' samples. Only this target biomarker HP-1 showed a well discriminated patterns between cancer and control subjects totally about 30-40 subjects. In addition, we have spent almost 36 months to collect samples, typically patients of T3-4 advanced oral cancer in the hospital. Therefore, the preliminary results gave us encouragement to publish urgently. Then, we filed this development of SPR method to US patent. Now we hope to publish this study in this journal. So we decided sample size as 30-40 subjects in this study empirically. The further confirmation study will precede large sample size numbers in the future.

The additional descriptions are in the section of material and methods, sentence #66-68, and in the section of discussion, sentence #224-232.

Thank for recommendations and appreciate your suggestions
